# Early Events after Herpes Simplex Virus-Type 1 Entry Are Necessary for the Release of Gamma-Hydroxybutyrate upon Acute Infection

**DOI:** 10.3390/ph16081104

**Published:** 2023-08-04

**Authors:** Faith O. Osinaga, Yu-Chih Chen, Madan K. Kharel, Yan Waguespack, Sichu Li, Shaochung Victor Hsia

**Affiliations:** 1Department of Pharmaceutical Science, School of Pharmacy, University of Maryland Eastern Shore, Princess Anne, MD 21853, USA; 2Department of Natural Science, School of Agriculture and Natural Science, University of Maryland Eastern Shore, Princess Anne, MD 21853, USA; 3Knowledge Bridge, LLC, Fairfax, VA 22032, USA

**Keywords:** gamma-hydroxybutyrate, UPLC-MRM-MS, virology, mass spectrometry

## Abstract

We reported that gamma-hydroxybutyrate (GHB) is released upon Herpes Simplex Virus Type-1 (HSV-1) acute infection. However, the cellular biochemical processes involved in the production of GHB in infected cells are unclear. This study aims to shed light on the biochemical pathway and the stage within the viral life cycle responsible for the release of GHB in infected cells. UV-inactivation, acyclovir (ACV), and cycloheximide (CHX) treatments were used to inhibit HSV-1 replication at various stages. Vero cells treated with UV-inactivated HSV-1 significantly decreased GHB production. However, ACV or CHX treatments did not affect GHB production. We also showed that inhibition of glycolytic enzyme enolase by sodium fluoride (NaF) significantly reduces GHB production upon infection. This finding suggests that suppression of glycolytic activity negatively affects cellular GHB production. Our data also indicated that succinic semialdehyde dehydrogenase, an enzyme involved in the shunt of the tricarboxylic acid (TCA) cycle to generate succinic acid, was decreased upon infection, suggesting that infection may trigger the accumulation of succinic semialdehyde, causing the production of GHB. Although the precise mechanism has yet to be defined, our results suggest that early events following infection modulates the release of GHB, which is generated through the metabolic pathways of glycolysis and TCA cycle.

## 1. Introduction

Herpes Simplex Encephalitis (HSE) is rare but quite lethal with a death rate of 70–90% if left untreated [1]. Furthermore, over 50% of patients that survive HSE experience neurological deficits [2]. Moreover, HSV-1 infection is the leading cause of corneal blindness in industrialized countries [3,4]. Innervation of Herpes Simplex Virus Type-1 (HSV-1) can establish latency and later undergo reactivation in response to various factors, such as stress, skin trauma, and cold temperature [5]. Successful infection depends on several critical steps initiated by the attachment to specific receptors on the host cell membrane, and viral entry [6]. The HSV-1 virion initiates its life cycle upon contact with host cells [6]. At first, the glycoprotein gB and glycoprotein gC on the viral envelope is recognized by the heparan sulfate proteoglycans (HSPGs) on the host cell membrane, and binding is achieved [7]. Further interactions between the viral glycoprotein gD and cellular receptors, such as nectin-1, nectin-2, herpes virus entry mediator (HVEM), and 3-O-sulfated heparan sulfate occur [8]. The N-terminal domain of gD binds to cellular receptors, leading to the exposure of the C-terminal domain and the subsequent recruitment and activation of gB and the gH/gL heterodimer, ultimately trigger membrane fusion. HSV-1 entry can occur either by direct fusion of the viral envelope with the host cell membrane or by an endocytic mechanism [9]. In fully permissive, non-neuronal cultured cells after entry, HSV-1 encoded around 90 unique transcriptional units and completed the entire life cycle in less than 20 h [7]. Previous studies have reported that herpesviruses’ viral replication is highly dependent on glucose and the tricarboxylic acid cycle [10].

The released volatile organic compounds (VOCs), upon a viral infection, such as influenza, adenovirus, respiratory syncytial virus, rhinovirus, and parainfluenza virus, have been suggested to be potential biomarkers for early diagnosis [11]. The mechanisms of VOC production in mammalian cells are not completely known and the intrinsic characteristics of the possible importance of VOC release upon viral infection merit investigation. We previously reported that HSV-1-infected cells produce gamma-hydroxybutyrate (GHB) as the key pathway intermediate for the subsequent production of a VOC gamma-butyrolactone (GBL) [12]. The GHB production was detected as early as 5 h post infection (hpi) using pre-column derivatization with 3-nitrophenylhydrazine (3-NPH). The 3-NPH derivatization allowed for a sensitive, selective, and reliable method for the indirect detection of GHB in cell media since GHB does not absorb UV [12]. We propose that GHB is part of the cellular self-defense mechanism against the virus attacking neighboring cells, similar to the bacteria stress response signaling mechanism, quorum-sensing [11]. The characterization of the biosynthetic pathway for the production of VOCs in mammalian cells has been elusive to date. Bioinformatics analyses have not revealed genes homologous to bacterial GBL homologous genes, suggesting the existence of distinct enzymes in the mammalian system.

Studies have shown that HSV-1 infection increases 6-phosphofructo-1-kinase (PFK-1) expression, which increases PFK-1 total activity, and ATP content [13]. PFK-1 is a key rate-limiting and regulatory enzyme within the glycolytic pathway. PFK-1 is highly regulated by allosteric modulators such as 2,6-bisphosphate, ATP, and citrate [13,14]. Many viruses can modulate glucose metabolism [10,14]. For instance, human cytomegalovirus (HCMV) activates glycolytic metabolism. Mechanistically, HCMV promotes CamKK-dependent activation of PFK-1 while the Mayaro virus elevates the level of 2,6-bisphosphate to enhance PFK-1 activity [14]. Moreover, it was also reported that inhibition of glycolysis regresses HSV replication and SARS-CoV-2 replication [14,15]. To keep central carbon metabolism elevated for HSV-1 nucleotide synthesis, the virus can modulate glutamine to feed into the tricarboxylic acid (TCA) cycle through the synthesis of alpha-ketoglutarate [14]. Subsequently, the amino acid, glutamine, can undergo deamination to form glutamate, which then serves as a precursor to GHB [14,16]. These phenomena align fairly well with the previous report that increased glycolysis may enhance endogenous GHB production in bacteria [17].

GHB metabolism is closely modulated by glycolysis. The mechanisms by which HSV-1 modulates glycolysis to increase central carbon metabolism and its ability to enhance GHB production by the regulation of glutamine metabolism merit investigation. Earlier studies showed that HSV-1 can affect glycolysis [13,14]; however, the mechanism by which HSV-1 triggers glycolysis to modulate GHB release has not been investigated. In this study, we investigated the various stages of the viral infection and their effect on glycolysis and GHB production. We used the indirect detection of GHB through its derivatization using 3-nitrophenyhydrazine (3-NPH) [12]. We found that the GHB production is released at the low multiplicity of infection (MOI) of 0.1 and is selective to HSV-1 infections. Early events after viral entry trigger cellular GHB production and the majority of GHB accumulates in the intracellular space. We found that both glycolysis and the TCA cycle modulate GHB release indirectly upon viral infection.

## 2. Results

### 2.1. HSV-1 Infection Triggers the Release of GHB

To test the release of GHB upon HSV-1 infections, Vero cells were infected at multiplicities of infection (MOI) of 0.5 and 1 with different HSV-1 strains followed by GHB detection via one-step derivatization using 3-NPH and MS analysis. The results showed GHB release was triggered upon HSV-1 KOS, Mckrae, and 17 Syn^+^ strains at MOI 0.5 and 1. A 1.4-fold increase in mean peak area for GHB-NPH derivative was observed in cells infected with 17 Syn^+^ at the MOI of 0.5 compared to the KOS and Mckrae strains. At the MOI of 1, the 17 Syn^+^ strain infection resulted in a 1.6- and 1.5-fold increase in GHB production compared to the KOS and Mckrae strains, respectively (Figure 1a). No significant difference in the amount of GHB production was observed at MOI 0.5 for KOS and Mckrae nor at the MOI of 1. When comparing GHB productions at different MOIs with the HSV-1 17 Syn^+^, the highest production of GHB was observed at an MOI of 1. GHB production in the infected cells was found ~25-folds more compared to non-infected cells (Figure 1b). Similarly, 7.1-fold more production of GHB was observed in infected cells at MOI 0.1. Interestingly, a ~22-fold increase in GHB production was observed at MOIs of 0.5 and 3 compared to the control sample. The differences in GHB production between MOIs of 0.5 and 3 were statistically insignificant (Figure 1b). 17 Syn^+^ at the MOI of 1 triggered the highest GHB production and was selected to be tested in the rest of the following experiments.

To assess the GHB release upon infection in Vero cells, GHB was measured using LC-MRM-MS, as described in Section 2.4. The results showed the release of GHB from both the media (supernatant) and cell pellet in samples infected for 24 h (Figure 2). The mean concentration of GHB in the media was 5.4 pg/mL. The mean intracellular concentration of GHB was determined to be 1.8 ± 0.02 pg/cell. Thus, 1.8 × 10^−6^ μg of GHB was found intracellularly compared to 3.4 × 10^−7^ μg in the media.

### 2.2. Viral Replication and Viral Protein Synthesis Are Not Required for GHB Production

The potential modulation of GHB production upon HSV-1 infection was further investigated by inhibiting various stages of the viral replication cycle. The antiviral agent acyclovir (ACV) was used to inhibit early-stage viral DNA synthesis. The treatment of culture with ACV (100 µM final concentration) resulted in no significant difference in GHB release compared to the HSV-1 infected samples (5 hpi and 10 hpi) without ACV treatment (Figure 3a). The results suggested the inhibition of viral replication does not affect the release of GHB. Next, viral protein synthesis was inhibited with 100 µg/mL (final concentration) of cycloheximide (CHX). We did not find a significant effect of CHX treatment on GHB productions when compared with untreated samples at 5 hpi and 10 hpi (Figure 3a). The potential delay for the onset of actions of CHX following the treatment was minimized by pretreating cells with the antibiotic. At 24 hpi, a marginal increase in GHB release of ~1.1- and ~1.3-fold was found in the samples treated with ACV and CHX, respectively, compared to the 10hpi treatments. A comparable increase in GHB production (~2.4-fold) was found in HSV-1 infection without treatment at 24 hpi compared to 5 hpi and 10 hpi, further indicating the insignificant effect of antibiotic treatments. To corroborate the results of the GFP labeling experiment, the release of infectious viral particles was tested by collecting the media of infected Vero culture, followed by plaque assay. As expected, infectious viral particles were not found in both the ACV- and the CHX-treated samples. The results show that a statistically significant (α = 0.05) amount of virus titers (approximately 5.1 × 10^5^ PFU/mL) was found in the control infected samples without antibiotics, whereas no plaques could be detected for infected cells treated with ACV and CHX or in the control samples without viral infections (Figure 3b). These results cumulatively indicate that viral replication and viral de novo protein synthesis does not modulate the release of GHB.

To ensure antibiotics (CHX and ACV) were inhibiting viral replications, viral replications were monitored using green-fluorescent-protein (GFP)-labeled HSV-1 17 Syn^+^ strain. Vero cells were infected with the labeled virus at MOI of 1 in the absence or the presence of ACV or CHX. At 24 hpi, viral titer was analyzed using qualitative fluorescence microscopy, which did not produce any signal in the presence of CHX, whereas a weak signal could be observed in the sample treated with ACV. These results clearly show that the antiviral agents were inhibiting viral replication as expected, and the lack of significant effect on GHB production was unrelated to viral inhibitions (Figure 4).

### 2.3. Viral Entry Decreases GHB Production upon Infection

Previously, we showed that GHB release occurs as early as 5 hpi [12]. Therefore, we hypothesize that the early stages of the viral replication cycle trigger a cellular response by releasing GHB. To test this hypothesis, we performed infections using a UV-inactivated virus. The UV-inactivated viruses are capable of penetrating the cells but are incapable of progressing through the replication cycle including early-stage transcription, translation, and replication. Interestingly, the infection of cells with UV-inactivated virus did not result in significant GHB release at 10 hpi and 24 hpi compared to the regular virus (Figure 5). This suggests that the entrance of the virus into the cells alone is not sufficient for GHB production. GHB production is likely triggered by post-entry viral activities.

### 2.4. Inhibition of Glycolysis Decreases GHB Release upon Infection and Viral Replication

It is well established that HIV, HCV, and HCMV affect cellular metabolism, such as glycolysis [14]. An earlier study reported that HSV-1 activates glycolysis by 6-phosphofructo-1-kinase (PFK-1) and the inhibition of glycolysis negatively affects HSV-1 replication [13]. To test whether glycolysis mediates the production of GHB, Vero cells were infected with HSV-1 and treated with an enolase inhibitor, sodium fluoride (NaF). The results show that infected cells treated with NaF decreased GHB production by ~1.6-fold (α = 0.05) compared to the control infection sample without treatment at 24 h (Figure 6a). Decreased GHB release upon treatment of NaF paralleled the reduction of viral GFP signal and viral plaques when observed by plaque assays (Figure 6b), corroborating the previous reports on the negative impact of suppressed glycolysis on viral replication [13].

### 2.5. Impact of HSV-1 Infection on GHB-Metabolic Enzymes

Several enzymes are implicated in the metabolism of GHB [18,19]. Glutamine is a well-known precursor to GHB, which requires three enzymatic conversions catalyzed by glutamic acid decarboxylase (GAD), GABA transaminase (GABA-T), and GHB dehydrogenase (GHB-DH) as shown in Figure 7a [19]. An alternative route for the production of GHB is reported to be from rat livers via hydroxyacid-oxoacid transhydrogenase (HOT), which converts SSA to GHB through a coupled metabolism of α-ketoglutarate to D-2-hydroxyglutarate. Its gene, ADHFel, is highly correlated with bacterial GHB dehydrogenase and free of NAD- or NADP-binding sites [18]. To assess the impact of viral infections on the expression of the genes corresponding to these metabolic enzymes, we conducted comparative gene expression studies using RT-qPCR. The results indicated that SSA-DH was significantly decreased during the HSV-1 infection (Figure 7b). SSA-DH participated in the anabolism of succinate via the γ-aminobutyric acid (GABA) shunt (Figure 7a). It is likely that SSA-DH reduction prompted the accumulation of succinic semialdehyde (SSA), pushing for the production of GHB upon infection (Figure 7b).

## 3. Discussion

Our previous reports showed that a VOC GBL was released upon acute HSV-1 infection of Vero cells [20]. We also reported that the spiked GBL production is due to an enhanced production of gamma-hydroxybutyrate (GHB). The conversion of GHB to GBL could occur via an enzyme-catalyzed process or spontaneous lactonization of GHB [12]. To the best of our knowledge, GHB release triggered by infection of DNA virus is not known. Physiologically, the roles of GHB in humans have not been well established [21,22]. It is known that GHB is an important metabolic pathway intermediate and its cellular concentration is impacted by a variety of cellular and biochemical signals [23]. The most studied one involves the conversion of glutamic acid to GHB via GABA (Figure 6a). In addition to the physiological role of GHB in mammalian cells, the lactone form of GHB also serves as a stress signal for bacteria [17]. Due to the increase in GHB production at the early stage of the viral infection, we cannot rule out the possibility that GHB may play a role in the cellular self-defense mechanism against the virus. To provide more insights into how the virus modulates enhancement in the production of cellular GHB, we further examined GHB production during the early stages of HSV-1 replication.

Our findings clearly show that HSV-1 infection is needed for enhanced GHB production and release. The extent of GHB production varies significantly with different strains, the most abundant being in the 17 Syn^+^ strain (Figure 1a). It is noteworthy that 17 Syn^+^ is a robust HSV-1 strain used routinely to conduct infections in in vitro cellular models [24,25]. The increased production of GHB in the infection sample, compared to non-infection, is statistically significant from a low multiplicity of infection such as 0.1 to the most abundant at 1 (Figure 1b). This clearly shows that the GHB release observed is in response to the HSV-1 infection. The extracellular GHB found in the infection samples corresponded to the intracellular concentrations, suggesting the cellular production of GHB and release. It is unclear whether the released GHB serves as a biochemical signal to elicit a cellular response in neighboring cells.

To detect which stages within the viral cycle can induce GHB upon infection, viral entry, viral replication, and viral protein expression were inhibited by a UV-inactivated virus, ACV, and CHX, respectively. Our results showed that the UV-inactivated virus treatment did not result in GHB release at 10 hpi and 24 hpi (Figure 4). Neither the increased duration of infection nor the titer used for the infection could trigger any GHB release when using the UV-inactivated strain. This provides a clear evidence that the entry of the virus alone is not capable of triggering cellular response leading to a GHB production, this rather requires downstream events. Interestingly, the GHB production was not affected by the inhibition of viral protein synthesis or viral DNA replication, ruling out the significance of these major early viral replication events on the GHB production.

Glycolysis and the TCA cycle are the central metabolic pathways in all organisms. Modulation of these pathways upon cellular stress is expected [16]. HSV-1 enhances glycolysis by elevating the activity of the glycolytic pathway enzyme, 6-phosphofructo-1-kinase (PFK-1). Consequently, the inhibition of glycolysis also affects HSV-1 replication by limiting the production of ATP, which is crucial for viral replication [14]. GHB is derived from L- glutamic acid and is found in various tissues and fluids such as rat brains, human plasma, and urine specimens [21,22]. Glutamine can convert into succinic semi-aldehyde through the intermediate gamma-aminobutyric acid (GABA)-a well-known neurotransmitter [23]. SSA can undergo either oxidation or reduction in the subsequent metabolic routes. Oxidation of SSA by succinic semi-aldehyde dehydrogenase (SSADH) generates TCA intermediate succinic acid (Figure 6a). The reduction of the SSA by succinate semi-aldehyde reductase (SSAR) results in the production of GHB, whereas the reverse reaction is catalyzed by GHB dehydrogenase [21,22,23]. Since GHB metabolism is integrated with the TCA cycle [23]; and the HSV-1 replication is positively correlated with glycolytic activities [13], we investigated the impact of glycolytic activities on the GHB release in HSV-1 infected cells.

Inhibition of enolase activities by NaF significantly decreased the GHB production at 24 hpi. This suggests that the cellular GHB production is negatively affected by the suppression of glycolytic activities. Such suppression would produce a negative impact on TCA cycle activities. As a result, alpha-ketoglutarate (a TCA cycle intermediate) to L-glutamine (a precursor for GHB) conversion could be suppressed (Figure 7). The results indicate a positive correlation between the GHB production and glycolytic/TCA cycle activities. However, the exact underlying mechanism leading to this observation is yet to be determined. We observed extremely low levels of expression of all of the tested glutamine metabolic enzymes (SSADH, GHBDH, GABAT, and SSAR) in infected cells compared to non-infected cells. However, the GHB production was elevated in the infected cells. We hypothesize that infection elevates both glycolytic and TCA cycle activities and minimizes the metabolic flux towards the TCA cycle via succinic semi-aldehyde. As a result, a majority of the SSA could convert into GHB, rather than succinic acid. The significance of the elevated expression of SSADH compared to other glutamine-metabolic enzymes in infected cells is yet to be determined. Based on the results, we suggest that the elevated GHB production upon HSV-1 infection in epithelial cells is caused by the modulation of the glutamine metabolic pathway, likely through the redirection of metabolic flux (succinic semi-aldehyde to GHB). This could be due to the elevation of glycolytic and TCA cycle activities caused by HSV-1 infections, which potentially minimizes a need for energy production through succinic semi-aldehyde to succinic acid conversion. Cumulatively, the present work offers new insights into biochemical processes involved in an enhanced cellular GHB production upon HSV-1 infection.

## 4. Materials and Methods

### 4.1. Cell Line and Virus

African green Monkey kidney cells Vero cells were purchased from ATCC (Cat #: ATCC CCL-81). They were grown in Modified Eagle Medium (MEM) containing fetal bovine serum (10%) and Antibiotic-Antimycotic (1%). The HSV-1 strain 17-Syn^+^/GFP, KOS, and Mckrae-GFP were all gifts from Dr. Gus Kousoulas (Louisiana State University, Baton Rouge, LA, USA) and used for the infection. The infection was performed on 35 mm cell culture plates (1.2 × 10^6^ cells/plate) at various multiplicity of infection (MOI) of 0.1, 0.5, 1, 2, and 3. Amid infection, the inoculums were treated with viral inhibitor drugs: acyclovir (ACV), cycloheximide (CHX), or sodium fluoride (NaF) either at pre- or post-infection. Infected cells were collected followed by freeze/thaw 3-time to break open the cell wall. Samples were mixed 10 times and subjected to derivatization as previously described [12]. To quantify GHB secretion into cell media and intracellular production, first, the cultured media and cell pellets were harvested separately from 35 mm Nunc cell culture and then subjected to one-step derivatization. To harvest the cell pellet, lysis cells underwent 3 cycles of freeze/thaw before being subjected to one-step derivatization.

### 4.2. Viral Infection and Viral Inhibition

For HSV-1 infection, Vero cells were pretreated with 100 µM Acyclovir (Cat #: A4669; Sigma-Aldrich, St. Louis, MO, USA) for 1 h, followed by infection. Vero cells were treated with 100 µg/mL Cycloheximide (Cat #: 66819, Acros Organics, Geel, Belgium) post-infection to inhibit viral de novo protein synthesis and viral proliferation, respectively. Vero cells were post-treated with 10 mM of NaF (Cat #: 470302-536, Avantor, Radnor Township, PA, USA) to prevent glycolysis. To examine viral entry without transcription and translation, Vero cells were infected with an ultraviolet (UV)-inactivated virus. Spectroline UV Crosslinker Select Series (Model XLE-1000, No. 1010-1) operating on the optimal crosslink setting with the 17-Syn^+^/GFP virus was used to prepare UV-inactivated viruses. Similar to infected samples, UV-inactivated virus was prepared at a multiplicity of infection of 1 and harvested at 10 and 24 hpi.

### 4.3. Preparation of Standards and Internal Standard (IS)

A stock solution of GHB (100 μM) was in the complete Vero medium prepared by diluting GHB from Sigma-Aldrich (St. Louis, MO, USA) (Cat#: G-001). The process of derivatization was described previously [12]. GBL (Cat#: B103608, Sigma-Aldrich, St. Louis, MO, USA) was used as the internal standard.

### 4.4. Pre-Column Derivatization of GHB in Cell Media

The pre-column derivatization was performed as described previously [12]. An amount of 64 mg of 3-nitrophenylhydrazine hydrochloride (3NPH·HCL) (Cat#: 642983, Sigma Aldrich St. Louis, MO, USA) was dissolved in 80% ethanol to prepare 64 mM solution. An amount of 240 mg of 1-ethyl-3-(3-dimethylaminopropyl) carbodiimide (EDC·HCL) (Cat#: 2595538, Alfa Aesar, Tewksbury, MA, USA) was dissolved in ultrapure water to prepare 310 mM solution. Pyridine (Cat #: 270970, Sigma Aldrich, St. Louis, MO, USA) was added to this solution to produce (EDC·HCL) (98.5%)/ pyridine mixture (1.5%) (*v*/*v*) following the previously reported protocol with slight modifications [26,27]. For the derivatization, each reagent was prepared freshly for each solution. An amount of 400 μL of 3-NPH solution (64 mM) was added into 400 μL of the standard solutions of GHB or cell culture sample solutions in microcentrifuge tubes. Then, 800 μL of EDC HCL/pyridine solution was aliquoted into the mixture. The sample was vortexed for 30 s. Next, the mixture was incubated at 60 °C for 30 min. The reaction samples and standards solutions were centrifuged at 1500× *g* for 5 min. Then, 250 μL of the reaction sample was diluted with 750 μL of ultrapure water and supplemented with 110 μL of GBL (0.3 μg/mL). The diluted sample supplemented with the internal standard was mixed using a pipette. The GHB-NPH derivatized product is not commercially available; therefore, an indirect detection method was developed to quantify GHB. An amount of 4 μL of each reaction samples (standards, quality control, and culture extracts) was injected in the UPLC-MRM/MS for analysis. All data files were processed and statistically analyzed with Mass Lynx software (version 4.2). Triplicate injections were employed for each of the samples, quality control, and standardized at each concentration to collect triplet data points (*n* = 3). The pre-column derivatization and analysis was performed as described previously [12].

### 4.5. UPLC/MRM-MS

As described previously [12], the LC separation was performed on the Waters binary UPLC system (Acquity H-class) in the reverse phase using a C18 silica column (XSelect HSS T3 100 Å, 2.5 μm, 4.6 × 100 mm; P/N: 186006262) assembled with a guard column (Waters VanGuard Cartridge Holder). The gradient comprised binary solvents (solvent A: 0.1% (*v/v*) formic acid in water; solvent B: 0.1% (*v/v*) formic acid in acetonitrile) and was used throughout the duration of the experiment. The flow rate was 1.0 mL/min from 0 to 9.6 min. The total run time was set to 9.6 min. The binary gradient consisted of 92% A and 8% B from 0 to 4.1 min; 60% A and 40% B from 4.1 to 4.5 min; then 92% A and 8% B from 4.6–9.6 min. The temperatures of column chamber and autosampler were maintained at 32 °C and 25 °C, respectively, since the derivative is stable at room temperature [12]. To increase the selectivity and sensitivity of the GHB NPH derivative, multiple reaction monitoring (MRM) scans were used for analysis. The fragments of the GHB NPH derivative were identified using direct infusion to the MS. The fragment with the highest intensity (*m/z* = 86.8) was used to analyze the MRM channel for standard, quality control, and sample runs and was also utilized to monitor the ion of the internal standard (GBL). The GHB NPH derivative identified with *m/z* = 86.8 was produced in MRM scans and was consistently used for all of the quantitative experiments.

GBL, as the internal standard was monitored at [MH]+ ion peak (*m/z* = 86.8) as the precursor ion and the product ion at [MH]+ ion (*m/z* = 44.9) from single-ion recordings (SIR) and MRM scans, respectively. The product ion peak (*m/z* = 44.9) is a ubiquitous fragment of GBL in electrospray ionization (ESI) scans in the positive mode (Table 1). Waters binary UPLC system (Acquity H-class) coupled with the Xevo TQ-S Micro was equipped with an ESI source for MS analysis. The MS instrument was operated in the positive mode for MRM scans to quantitatively measure the GHB NPH derivative. The GHB NPH derivative was identified at the cone voltage of 40 V for the [MH]+ ion 240.1 Da, (molecular weight of the derivative) and for the product peak [MH]+ ion (86.6 Da) (Table 1). The collision energy for the product ion [MH]+ was optimal at 15V. The optimal ESI operating parameters for the analysis of the GHB NPH derivative were the following: capillary voltage of 55 kV, desolvation temperature of 600 °C, cone gas flow at 1 L/h, source gas at 996 L/h, and argon was employed as the collision gas.

The derivative reaction is highly selective and it will only interact with carboxylic acids, which is only in the presence of GHB. No such derivatized product was detected in the presence of spiked GBL [12]. We could not monitor the GHB concentration directly as it would slowly convert into GBL during analysis. Therefore, the only viable alternative was to use pre-column derivatization to indirectly quantify GHB, then characterize the derivatized product using MRM mode, and monitor fragmentation products. We considered various internal standards that are chemically similar to GHB. We wanted to choose one that has similar functional groups, boiling points, and activity as GHB to preserve the precision and accuracy of this method. Previously, we first experimented with the use of GHB-d6 (C_4_HNaO_3_D_6_) as an internal standard for pre-column derivatization with 3-NPH. As a result, the generated linear curve had a lower correlation coefficient <0.999. The deuterated GHB did not react consistently with EDC, pyridine, and 3-NPH. We could not explain the reasons for such inconsistency. We then chose GBL as the internal standard to ensure the consistency of the spectrometer function. GBL is commercially available and it can be easily and reliably quantified when aliquoted to the controls, standards, and samples. With GBL as the internal standard, we were able to generate a sensitive method for the indirect quantitation of GHB. GHB derivative eluted at 3.49 min and GBL (internal standard) eluted at 3.14 min, indicating no overlap of their retention times. Since retention times for GHB derivative and GBL are different, a minimal interference of GBL on analysis can be expected. We ensured that the amount of GBL (internal standard, 0.3 μg/mL) was significantly higher by >1000-fold and consistent.

### 4.6. Real-Time qPCR

Viral and mammalian gene expression was quantified as described previously [28]. RNA was collected using the iScript sample preparation reagent (catalog no. 170-8898; Bio-Rad, Hercules, CA, USA), followed by reverse transcription using iScript RT Supermix (Cat #: 17008841; Bio-Rad, Hercules, CA, USA) to produce the cDNA template. The qPCRs were performed using SsoAdvanced Universal SYBR green supermix (Cat #: 1725271; Bio-Rad, Hercules, CA, USA) on triplicate samples with primers specific for the selected viral or mammalian genes listed in Table 2 [18]. The reverse transcription reaction was carried out at 50 °C for 10 min. The qPCRs were initially carried out at 95 °C for 1 min, followed by 39 cycles at 95 °C for 10 s, then 57 °C for the 30 s and 65 °C for 5 s. For each gene, mRNA expression was normalized by peptidylprolyl isomerase A expression (PPIA; forward, 5′-AGCATACGGGTCCTGGCATCT-3′; reverse, 5′-CATGCTTGCCATCCAACCACTCA-3′) [28].

### 4.7. Plaque Assay

The supernatants were collected from infected cells in 35 mm Nunc cell culture plates. The supernatants were serially diluted before being added to a monolayer of Vero cells in a 96-well plate seeded with 5 × 10^4^ cells/well. The inoculates were left on the plate for one hour for viral attachment and entry. The inoculums were then replaced with fresh culture media and incubated for 48 h. After 48 h, the cells in the 96-well plate were fixed with ice-cold methanol for 10 min and were left to dry for 1 h. Then, wells were stained with crystal violet. Each visible plaque was counted within each well and data were collected in triplicates.

### 4.8. Fluorescent Microscopy

Each fluorescent and bright field image were recorded with an Olympus fluorescence microscope (IX71) supplemented with an Olympus DP71 digital camera (Olympus America Inc., Center Valley, PA, USA). Exposure times were <1 s. Images were recorded for HSV-1-infected samples at the MOI of 0.1, 0.5, 1, and 3.

### 4.9. Data Analysis

The GHB NPH derivatization analysis was calculated as described previously [12]. Statistical analyses were performed using Student’s unpaired t test and one-way analysis of variance (ANOVA) in Excel. All data values are presented as mean ± standard errors of the mean (SEM), and a *p* value of <0.05 was considered statistically significant. The lower limit of detection (LLOD) was the lowest concentration that would yield a signal to noise ratio of at least 3 and have a coefficient of variance (CV) of <15% for injections (*n* = 3). Lower limit of quantification (LLOQ) was the lowest concentration to yield a signal to noise ratio of at least 10 and have a CV of <10% for the injections at (*n* = 3). The LLOD, LLOQ and CV values were determined by the standard range. The standard range of the GHB NPH derivative was 1–6 μg/mL to generate a calibration curve. The response factor was generated by the linear curve with a high correlation coefficient value of R^2^ = 0.9998, to determine the molar concentration of GHB. The intra-day precision was determined as the CV% of the known concentrations of derivatized GHB that were freshly prepared each day and injected in triplicate. The inter-day precision was analyzed and compared for three consecutive days to calculate the CV%. (Table 3).

## 5. Conclusions

The release of GHB from HSV-1 acute infection is very complex and the molecular mechanism has yet to be characterized. Our present findings suggest that early infection events played a key role in the release of GHB, which is, in part, generated through the metabolic pathways of glycolysis and the TCA cycle. Future studies using RNA-seq, immunoblot, and NMR spectroscopy for transcriptomic, proteomic, and metabolomics studies, respectively, are underway.

## Figures and Tables

**Figure 1 pharmaceuticals-16-01104-f001:**
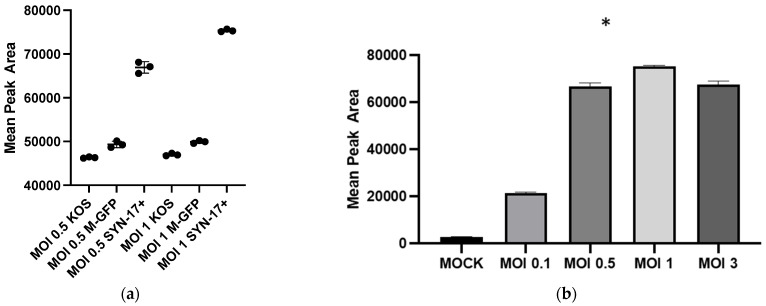
Mean peak area of GHB-NPH. Indirect detection of GHB by HSV-1-infected Vero cells (**a**). Samples infected with 17-Syn^+^, KOS, Mckrae at MOI 0.5 and 1 at 24 hpi. (**b**) Infected samples at MOI 0, 0.1, 0.5, 1, and 3 at 24 hpi. An (*), represents a significant statistical difference (*p* < 0.05) of GHB production compared (**a**) to samples at the same MOI and compared (**b**) to non-infection mock.

**Figure 2 pharmaceuticals-16-01104-f002:**
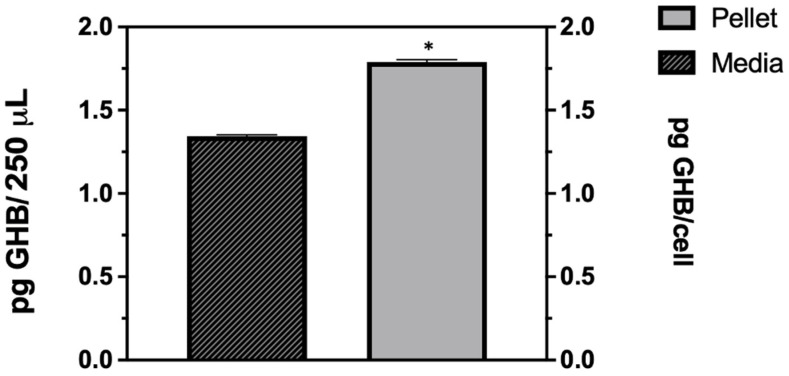
HSV-1 infection triggers the release of GHB in Vero cells. Quantitation of cellular GHB was measured intracellularly and within cultured media. Cells and 250 µL media were harvested from 35 mm Nunc cell culture plates, separately. Cells were subjected to lysis and then subjected to UPLC-MRM/MS analysis. Assays were performed in triplicates (*n* = 3) and means are shown with SD as error bars. Thus, 1.8 × 10^−6^ μg of GHB was found intracellularly compared to 3.4 × 10^−7^ μg in the media. Statistical analyses were performed with Student’s unpaired *t* test. An asterisk (*), represents the significance of GHB production between the infected samples, *p* < 0.05.

**Figure 3 pharmaceuticals-16-01104-f003:**
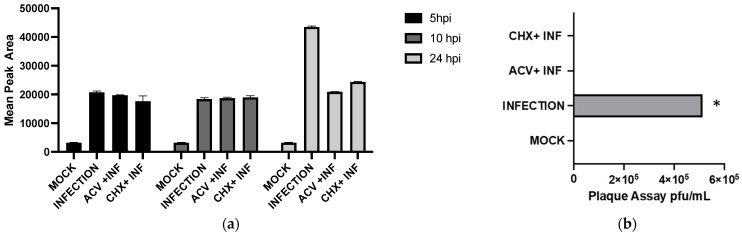
GHB production upon infection with different treatments to inhibit stages within the viral life cycle. Detection of GHB induction upon viral infection at 5, 10, 24 hpi with Acyclovir (ACV) and Cycloheximide (CHX) compared to infection and non-infection (mock). Infections were performed at MOI of 1 (**a**). Plaque assays were performed using lysates from infected Vero cells. Shown are the results of the untreated lysates, infection only, and treated with ACV and CHX. An asterisk (*) indicates a significant statistical difference (*p* < 0.05) compared to the non-infection mock sample (**b**).

**Figure 4 pharmaceuticals-16-01104-f004:**
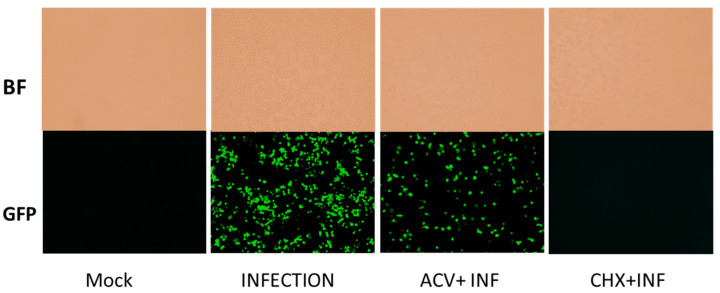
Bright field and fluorescent microscopy images for mock (non-infected) and infected samples at 24 hpi with ACV and CHX. Fluorescence microscopy records more cells with stronger green fluorescence, suggesting that ACV and CHX treatments in cells disrupted viral gene expression and replication.

**Figure 5 pharmaceuticals-16-01104-f005:**
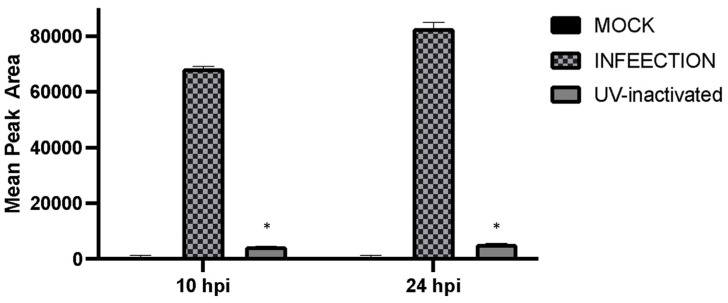
Detection of GHB was not seen in Vero cells infected by UV-inactivated HSV-1. Cells were infected with 17 Syn^+^ or UV-inactivated 17 Syn^+^ at MOI of 1.0 for 10 and 24 hpi. An asterisk (*) indicates a significant statistical difference (*p* < 0.05) compared with infection.

**Figure 6 pharmaceuticals-16-01104-f006:**
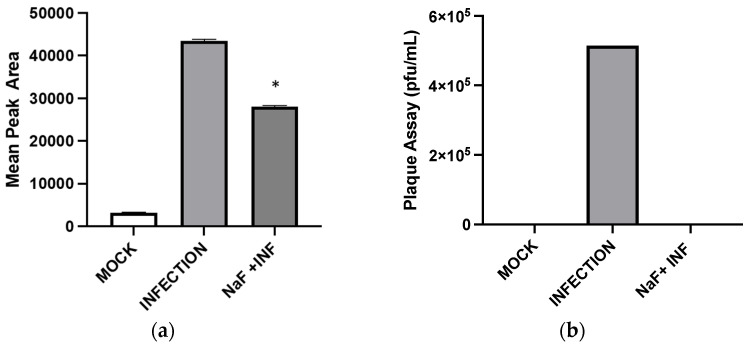
GHB production upon infection with NaF inhibits glycolysis. Detection of GHB upon viral infection at 24 hpi treated with NaF, infection, and non-infection mock (**a**). Plaque assays were performed using lysates from infected Vero cells. Shown are the results of the untreated, infected, and NaF-infected lysates. An asterisk (*) indicates a significant statistical difference (*p* < 0.05) compared to the non-infection mock (**b**).

**Figure 7 pharmaceuticals-16-01104-f007:**
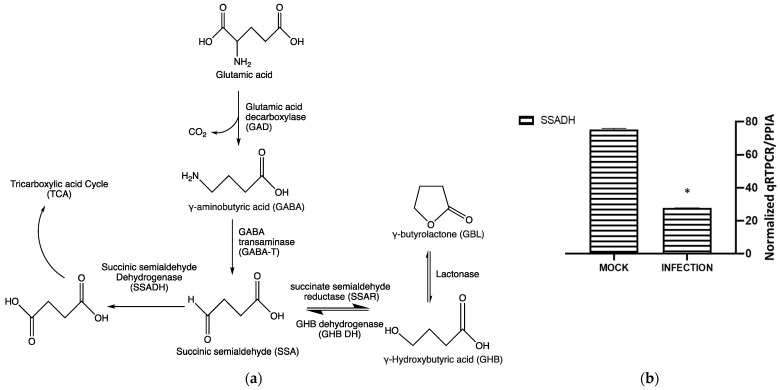
The production of GHB. (**a**) The pathway of GHB metabolism. (**b**) The glycolytic enzyme, SSADH expression was analyzed via RT-qPCR. Shown are the results of non-infection and infection. An asterisk (*) indicates a significant statistical difference (*p* < 0.05) compared to the expression level of other GHB metabolic pathway enzymes upon infection.

**Table 1 pharmaceuticals-16-01104-t001:** The precursor and product ions of derivatized GHB and internal standard GBL, respectively, with specific retention time, cone voltage, and collision energy. Ions produced in positive MRM mode were consistently used for all quantitative experiments.

		Retention Time (mins)	Precursor (*m*/*z*)	Cone Voltage (V)	Product (Da)	Collision Energy (V)	Ion Mode
GHB-NPH derivative		3.49	240.1	40	86.6	15	ES+
GBL		3.14	86.8	40	44.9	36	ES+

**Table 2 pharmaceuticals-16-01104-t002:** Genes, primers used for RT-qPCR.Succinate semialdehyde reductase (SSAR), GHB dehydrogenase (GHB-DH) succinic semialdehyde dehydrogenase (SSADH), GABA transaminase (GABA-T), and hydroxyacid-oxoacid transhydrogenase (HOT).

Gene (Enzyme)		Primers
*AKR7A2* (SSAR)	FWD-	5′-CAGGAATCGCTACTGGAAGG-3′
	REV-	5′-AAGTTCTGCTCCAGCTGCTC-3′
*AKR1A1* (GHB-DH)	FWD-	5′-TGCTGCTATCTACGGCAATG-3′
	REV-	5′-TGCATCAGGTACAGGTCCAG-3′
*ALDH5A1* (SSADH)	FWD-	5′-AGACCATCCTGGCTAACACG-3′
	REV-	5′-GGAGTCTCGCTCTGTCATCC-3′
*GABA-T*	FWD-	5′-TTCCACTCTTCCGCAGACTT-3′
	REV-	5′-GGGAGGCATACATCACCACT-3′
*ADHFe1* (HOT)	FWD-	5′-CAAGTAGCTATGGATTCCC-3′
	REV-	5′-GGTAGTGGCACTGCAATC-3′

**Table 3 pharmaceuticals-16-01104-t003:** Intra- and inter-day reproducibility of derivatized GHB measurements.

	Intra-Day (*n* = 3)		Inter-Day (*n* = 3)	
Gamma-Hydroxybutyrate Concentrations (μg/mL)	Mean Peak Concentration Mean ± SD	CV%	Mean Peak Concentration Mean ± SD	CV%
1	1.0842 ± 0.0119	1.1909	1.1045 ± 0.0943	9.4347
2	2.0640 ± 0.0707	3.5334	2.0345 ± 0.0478	2.3902
3	2.8238 ± 0.0090	0.3004	2.8135 ± 0.0341	1.1378
4	3.9313 ± 0.0144	0.3594	3.9499 ± 0.1583	3.9569
5	5.1759 ± 0.0406	0.9019	5.2079 ± 0.2007	4.0143
6	5.9771 ± 0.0124	1.0146	6.0118 ± 0.2845	7.1120

## Data Availability

Data is contained within the article.

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
