# Peer review of "Early Events after Herpes Simplex Virus-Type 1 Entry Are Necessary for the Release of Gamma-Hydroxybutyrate upon Acute Infection"

_pharmaceuticals, 2023, doi:10.3390/ph16081104_

Round 1

Reviewer 1 Report

The author reported that GHB is released upon HSV-1 acute infection and tried to investigate the cellular biochemical processes involved in the production of GHB in host cells. The author claimed that inhibition of viral DNA replication has no effect on GHB production, whereas the inhibition of glycolytic enzyme enolase significantly reduces GHB production upon infection with solid evidence. However, little data was provided to validate that viral infection may trigger the accumulation of succinic semialdehyde to cause the generation of GHB. More evidence should be offered: 1) The RNA-qPCR result with optimization, 2) Immunoblotting analysis of those proteins in the metabolic pathway, and 3) it would be better to perform RNA-seq or proteomics analysis with temporal resolution upon viral infection.

With some typos

Author Response

We agree with the reviewer that this is an interesting observation and more evidence is helpful. In addition to what the reviewer suggested such as western blot and RNA-seq, we are developing NMR spectroscopy for a metabolomics study and the preliminary results are promising. We plan to report these data in our next manuscript. In the meantime, we answer and describe the process in “Conclusion” from Line 460-462 to address the comment. In addition, we received help from an expert to edit the language.

Reviewer 2 Report

Nel manoscritto “Early Events after Herpes Simplex Virus- Type 1 Entry Are 2 Necessary for the Release of Gamma-Hydroxybutyrate upon 3 Acute Infection” gli autori hanno analizzato il rilascio di gamma-idrossibutirrato (GHB) a seguito di infezione da herpes. Lo scopo di questo lavoro è finalizzato ad analizzare il percorso biochimico e la fase all'interno del ciclo di vita virale responsabile del rilascio di GHB nelle cellule infette. Le cellule infettate sono state sottoposte a diversi trattamenti di inattivazione, dimostrando che i primi eventi post-infezione modulano il rilascio di GHB, che si genera attraverso le vie metaboliche della glicolisi e il ciclo del TCA.

The work was written quite well. There are some minor suggestions: 

-        In the introduction section, I would suggest inserting a few words about the mechanism of HSV-1 replication and the function of viral glycoproteins, which are important in promoting entry and viral replication. I would suggest reading a chapter written by Stelitano et al (HSV membrane glycoproteins, their function in viral entry and their use in vaccine studies; https://doi.org/10.1039/9781788013857-00014)

-        Please, rephrase the sentence in lines 238-240 “Considering the spike in GHB production upon HSV-1 infections at an early stage of 238 the replication cycle, the possible role of GHB in the cellular self-defense mechanism 239 against the virus cannot be ruled out.”; it’s not clear. 

-         Alla riga 304 sostituire “ATCC-CC-81” con “ATCC CCL-81” 

-         In alcune parti del testo sono presenti simboli errati, come ad esempio nella riga 334 (3NPH @HCL); linea 336 (EDC @HCL); linea 345 (400 @L) e (60@C) e altri.

Author Response

Reviewer 2

Nel manoscritto “Early Events after Herpes Simplex Virus- Type 1 Entry Are 2 Necessary for the Release of Gamma-Hydroxybutyrate upon 3 Acute Infection” gli autori hanno analizzato il rilascio di gamma-idrossibutirrato (GHB) a seguito di infezione da herpes. Lo scopo di questo lavoro è finalizzato ad analizzare il percorso biochimico e la fase all'interno del ciclo di vita virale responsabile del rilascio di GHB nelle cellule infette. Le cellule infettate sono state sottoposte a diversi trattamenti di inattivazione, dimostrando che i primi eventi post-infezione modulano il rilascio di GHB, che si genera attraverso le vie metaboliche della glicolisi e il ciclo del TCA. Response: We appreciate the reviewer’s comment.

-       In the introduction section, I would suggest inserting a few words about the mechanism of HSV-1 replication and the function of viral glycoproteins, which are important in promoting entry and viral replication. I would suggest reading a chapter written by Stelitano et al (HSV membrane glycoproteins, their function in viral entry and their use in vaccine studies; https://doi.org/10.1039/9781788013857-00014) Response: We concur with the reviewer that it is helpful to add more information about the roles of glycoproteins during the HSV-1 replication. We found the book chapter very useful and utilized some of the knowledge in the section of Introduction. The information is described from Line 36-45 (Highlighted) and the Book Chapter is now cited as one of the references.

-        Please, rephrase the sentence in lines 238-240 “Considering the spike in GHB production upon HSV-1 infections at an early stage of the replication cycle, the possible role of GHB in the cellular self-defense mechanism against the virus cannot be ruled out.”; it’s not clear. Response: We understand the concern. Therefore we changed the sentence to “Due to the increase of GHB production at the early stage of the viral infection, we cannot rule out the possibility that GHB may play a role in the cellular self-defense mechanism against the virus”. We believe it is straightforward. It can be found from Line 248-250.

-         Alla riga 304 sostituire “ATCC-CC-81” con “ATCC CCL-81”. On line 304 replace “ATCC-CC-81” with “ATCC CCL-81” Response: We thank the reviewer and made the correction. The correction is Line 314-315.

-         In alcune parti del testo sono presenti simboli errati, come ad esempio nella riga 334 (3NPH @HCL); linea 336 (EDC @HCL); linea 345 (400 @L) e (60@C) e altri. In some parts of the text there are incorrect symbols, such as in line 334 (3NPH @HCL); line 336 (EDC @HCL); line 345 (400 @L) and (60@C) and others. Response: Yes we corrected the symbols. The correction can be found scattered from Line 346-356.

Round 2

Reviewer 1 Report

Lines 460-462, described the funding information instead of the problem addressing process. 

Minor editing of English language required

Author Response

We appreciate the reviewer's comment but I'm not sure I understood it. The Line 460-462 was offered as a future direction and a response to the previous critique. The funding information is provided at Line 472-475.